# Modified Breath Figure Methods for the Pore-Selective Functionalization of Honeycomb-Patterned Porous Polymer Films

**DOI:** 10.3390/nano12071055

**Published:** 2022-03-24

**Authors:** Shahkar Falak, Bokyoung Shin, Dosung Huh

**Affiliations:** Department of Chemistry and Nano Science and Engineering, Center of Nano Manufacturing, Inje University, Gimhae-si 50834, Korea; sfalak1723@gmail.com (S.F.); sinbbo715@gmail.com (B.S.)

**Keywords:** self-assembly, modified breath figure method, interfacial reaction, honeycomb-patterned porous film, pore-selective functionalization

## Abstract

Recent developments in the field of the breath figure (BF) method have led to renewed interest from researchers in the pore-selective functionalization of honeycomb-patterned (HCP) films. The pore-selective functionalization of the HCP film gives unique properties to the film which can be used for specific applications such as protein recognition, catalysis, selective cell culturing, and drug delivery. There are several comprehensive reviews available for the pore-selective functionalization by the self-assembly process. However, considerable progress in preparation technologies and incorporation of new materials inside the pore surface for exact applications have emerged, thus warranting a review. In this review, we have focused on the pore-selective functionalization of the HCP films by the modified BF method, in which the self-assembly process is accompanied by an interfacial reaction. We review the importance of pore-selective functionalization, its applications, present limitations, and future perspectives.

## 1. Introduction

Highly-ordered patterned porous films mimicking natural honeycomb structures have attracted much research interest because of their ordered porosity, lightweight, stretchability, mechanical strength, and high surface area [1,2,3]. These properties make them highly desirable for potential applications such as antibacterial membranes [4,5], microelectronics [6], sensors [7,8,9,10,11], cell culture scaffolds [12,13], superhydrophobic surfaces [14,15], and size-selective separators [16]. These films can be obtained either via top-down lithographic techniques, such as photo-lithography [17] and contact lithography, or bottom-up fabrication such as self-assembly mechanisms of colloidal particles [18,19], emulsions [20,21], block copolymers [22,23], and phase separation [24,25]. Although the lithography process can reliably fabricate high-ordered porous films with a constant pattern, they usually require multiple processing steps and pre-patterned templates making the overall process tedious. Therefore bottom-up fabrication methods have attracted great interest from polymer and material scientists to obtain porous films with a high regularity due to their simple and facile fabrication process [26].

The bottom-up approach includes plasma arcing, chemical vapor deposition process, metal–organic decomposition, and self-assembly processes [27]. Among them, the self-assembly of water droplet arrays formed on cold substrates called the breath figure (BF) method has been utilized as a very simple and efficient technique for preparing films with highly-ordered pores, which are called honeycomb-patterned (HCP) films [28,29]. This is because the pores are hexagonally close-packed mimicking natural honeycomb structures. Widawski et al. first applied this method for the creation of the HCP porous polymer films by casting star polymer (polyparaphenylene) with polystyrene (PS) in carbon disulfide solution under a flow of moist air [30]. The BF procedure is not difficult, but the exact mechanism of the fabrication process is believed to be very complex because the films are obtained by a pure self-organization process carried out by the non-equilibrium state [1,2,31,32]. At present, a generally accepted mechanism for the fabrication of porous HCP patterned films by the BF is as follows: (a) polymer solution in a volatile solvent is cast under humid conditions; (b) evaporation occurs leading to condensation of water droplets onto the solution surface; (c) then the water droplets grow bigger, and sink into the solution; (d) the water droplets then pack because of Marangoni convection and thermocapillary effects, and gets stabilized at polymer/water interface via adsorption or deposition of the polymer to prevent coalescence; (e) they arrange into ordered two- or three-dimensional (2D or 3D) arrays; and finally, (f) a film with ordered porous structure is obtained after the solvent and water evaporate, as shown in Figure 1 [33,34]. The chemical functionality of the fabricated polymer films by the BF method remains unchanged and a porous pattern is obtained. However, controlling the hydrophobic/hydrophilic moiety in the polymer, use of highly volatile solvent and functional groups soluble in it remains a challenge. Perhaps the most serious limitation of this method is that the special chemical functionality of the HCP films is not possible by the BF method alone.

Researchers have greatly expanded and improved film-forming materials using various methods to overcome the above-mentioned challenges. There has been a great advancement in the BF method both in the film-forming materials and fabrication techniques. Researchers have used star polymers [30,35,36] to block and random copolymers [37,38,39,40,41,42], linear polymers with polar end groups [43,44], dendrimers with hyperbranched polymers [45,46,47], comb-like polymers [48], conjugated polymers [49,50], polymer blends [51,52], hybrids of polymers and nanoparticles (NPs), microgels or organic–inorganic polymer composites [53,54,55,56,57], the small molecules and supramolecular polymers, etc. [58,59], to impart functionality to the HCP films. The use of these polymers enables the fabrication of HCP porous films with multi-domain functionality in addition to the porosity. Additionally, the modification of polymer with fluorescing groups such as porphyrinated polyimide (PI) [60], platinum porphyrin-grafted poly (styrene-co-4-vinylpyridine) [61], or pyrene–PS mixture [7] HCP films was demonstrated to be useful in sensor applications due to their different fluorescing intensities under different environments. Therefore, the potential applications of the HCP films fabricated by the BF method have been greatly expanded in succession. It has been demonstrated that the HCP films are valuable in the fields of templating and microfabrication [62,63], biomaterials [64], superhydrophobic surfaces [65,66], sensing [60,67,68], catalysis [69,70,71], separation [72,73,74], responsive surface [75], coatings [76], surface-enhanced Raman scattering substrates [3,77], microchannels [78], and optical and conductive materials [79]. However, the above-mentioned functionalizing methods for the HCP films used either change homopolymer to copolymers, composites, or blend with the functional components. These methods require precise control of block length in the case of copolymers [5,80], or multistep laborious synthesis, as in the preparation of functionalized NPs [71]. A variety of functionalizing methods do exist from changing the parent polymer to copolymer or composite or blend [1,2,32,46,81].

In addition to the total surface functionalization of HCP film, recent developments in the field of the BF method have led to a renewed interest of researchers in the site-selective functionalization of HCP films at the top surfaces [82], bottom surfaces [83,84,85], or pore surfaces [86,87], as shown in Figure 2. The site-selective functionalization gives unique properties to the film which can be used for specific applications. The film develops a heterogenous surface by selectively functionalizing the top or the bottom surface only [82,83,84,85]. These heterogeneously functionalized films may have potential applications in the fields of bio-inspired artificial ion channels, directional drug delivery, biosensors, and molecular filtration [82,83,84,85]. For example, Guo et al. developed a unidirectional surface, where water could penetrate from the hydrophobic PI side to the hydrophilic anodic aluminum oxide side, but not from the hydrophilic to the hydrophobic side [85]. Among the three types of site-selective functionalization, the functionalization of pore surfaces is an active area of research, so we will focus on and review the pore-selective functionalization of the HCP films via the modified BF method. For a better understanding, we will introduce it by dividing it into two mechanisms: one by self-assembly alone without any chemical reactions during the BF process using block copolymer and Pickering emulsion method, the other by a modified BF process accompanying a chemical reaction during the BF process with various methods, called reactive breath figure (rBF). Among them, we focus more on the rBF method for the pore-selective functionalization of HCP porous films because the pore-selective functionalization by self-assembly has been reviewed in several reports [1,2,3,18,23,32,46,81].

## 2. Pore-Selective Functionalization of HCP Films

The functionality exclusive to the surface of the pores enables the use of these arrays in technical applications, such as the site-specific immobilization of proteins, essential to avoid hampering the activity of the protein which is beneficial to the specific selective adsorption of protein [43,88]. The protein orientation plays a critical role if protein microarrays are to be used as quantitative tools in biomedical research or clinically important protein-based nanomedicines [89]. For example, the surfaces prepared could be of potential interest as 3D cell culture platforms, that have been otherwise obtained via tedious multi-step procedures using lithographic techniques. The cell adhesion and proliferation on HCP films can be controlled by polymer film surface chemistry, which can be directed into the pores by functionalizing the pore-interior with glycopolymers or proteins [48,90,91]. This can enhance their proliferation and retain spherical morphology. Moreover, the specific patterning of metal or semiconductor materials could be applied in photonics, since the photonic band structure depends on the spatial periodic structures and the refractive indices of the dielectric materials in the structure [92].

### 2.1. Pore-Selective Functionalization of HCP Films by Self-Assembly

A great deal of previous research into pore-selective functionalization has focused on the self-assembly of block-co-polymers or amphiphilic materials, and Pickering emulsion of NPs [1,2,32,46]. Traditionally, pore-selective functionalization of HCP films is performed using a simple physical phenomenon, i.e., self-assembly of functional groups which are usually hydrophilic [93]. These involve physical phenomenon which was driven by the stabilization of interfacial tension at polymer solution/aqueous droplet interfaces [94]. If the film-forming polymer is amphiphilic or the polymer has a hydrophilic functional group in the hydrophobic polymer chain, the hydrophilic functional group will orient in the direction of the top surface of the film, to interact with humidity [81]. Therefore, the distribution probability of the hydrophilic group is higher at the top surface including the pore surface of the film while the hydrophobic polymer chain is highly distributed at the bottom surface of the film. The schematic representation of the self-assembly process demonstrating the possible mechanism is shown in Figure 3.

An innovative work was performed by Boker et al. by combining self-assembly and Pickering emulsions [86]. They were able to functionalize tri-n-octylphosphine oxide (TOPO) modified cadmium selenide (CdSe) NPs inside the pores of the HCP film. TOPO was used to stabilize the CdSe NPs and self-assemble around the interface of the water droplet/polymer solution during the BF self-assembly process, as shown in Figure 4. This method opens a simple route for the pore-selective immobilization of ligands or proteins having a wide variety of functional groups, curated for targeted applications in sensors, separation, or catalysis. Min et al. succeeded in the pore-selective functionalization of poly (acrylic acid) (PAA) in the PS HCP film by using a block copolymer of PS-b-PAA [87]. The block copolymer self-assembled around the water droplet during the BF process and the carboxylic group arranged inside the pores due to its amphiphilic property. The carboxylic acid (–COOH) groups inside the pores bonded with the hydrazide group of the biotin derivative for the patterning of proteins inside the pores, as shown in Figure 5. By the functionalization of PAA inside the pores, a range of other chemistries can be applied to pore-selectively immobilize proteins or other molecules of interest. For example, using this technique, Liang et al. successfully fabricated glucose-responsive HCP film used for self-regulated and controlled insulin release, as shown in Figure 6 [95]. The use of PAA to functionalize pores holds a lot of potential as multi-functional drugs, such as antimicrobial or anti-inflammatory molecules, could also be loaded onto the porous HCP film for direct drug delivery via skin [95]. Rodríguez-Hernández and co-workers effectively prepared antibacterial and antifouling materials selectively functionalized at the pore cavity for selective cell culturing [5,80]. They used blends of a diblock copolymer in a homopolymer matrix to precisely control the chemical functional groups inside the pores to obtain a selective surface for adhesion to mammalian cells while preventing bacterial contamination as shown in Figure 7. These methods used reverse microemulsion technique [95], or blends of a homopolymer (i.e., PS) and a block copolymer (polystyrene-b-poly (dimethylaminoethyl methacrylate) [5] which demonstrated the effective functionalization of pores in HCP polymer films. However, the experimental conditions and concentration of amphiphilic materials or NPs should be carefully controlled for pore-specific functionalization. For instance, the use of block copolymers such as PS-b-PAA [87], poly N-dodecylacrylamide-co-N-isopropylacrylamide) [96] needs precise length control of the block monomers to control the surface wettability, and further functionalization with bioactive molecules [87] or for NP deposition [86].

### 2.2. Pore-Functionalization by Self-Assembly Followed by Additional Treatment

Recent developments in the BF method have led to the advancement of the technique by self-assembly followed by additional treatment due to the possibility of combining organic and inorganic compounds. Therefore, the creation of new materials with tailored properties has attracted much attention and considerable literature has grown up around the fabrication of such materials via the modified BF method, summarized in Table 1 [70,97,98]. This method usually requires two steps for the pore-selective functionalization of the HCP films. Firstly, the polymeric porous film containing one reactant is fabricated under aq. humidity via the BF method. Secondly, a counter reactant or other physical agent is used to functionalize the pores of the HCP films. This method is generally popular for the synthesis of metal NPs, where a metal precursor is added in the polymer solution followed by its chemical reduction using a reducing agent such as sodium borohydride, sodium citrate, citric acid, ascorbic acid, and hydrazine [99]. For instance, Li et al. reported the addition of metal salts, such as HAuCl_4_, AgNO_3_, and CuCl_2_, in the polymer solution to fabricate metal-salt-filled HCP films [97]. Later, aqueous NaBH_4_ was poured onto the prepared films to reduce the metal salts inside the pores to metal NPs filled pores shown in Figure 8. Similarly, Modigunta et al. fabricated HCP porous PS films with an aldehyde group with functionalized pores using synthesized formylated PS (PS-CHO) [98]. The incorporation of the hydrophilic aldehyde group affected the hydrophobicity of the PS solution and assisted the self-assembly of PS-CHO towards the pore. The fabricated PS/PS-CHO film was then reacted with Tollen’s reagent to get Ag functionalized PS/PS-CHO at the pores, shown in Figure 9. Some researchers have utilized physical agents such as heat or UV for the functionalization of the desired material into the pores of the HCP porous films. In this technique, the reactive components are added to the polymer solution and made to react with the help of an external agent. In most recent studies, De León et al. fabricated hybrid membranes consisting of gold NPs (AuNPs) which were embedded in the pores of the PS HCP film [70]. They fabricated PS HCP films adding different concentrations of potassium tetrachloroaurate (III) (KAuCl_4_) via the BF method under aq. humidity. The KAuCl_4_ moved inside the pores of the membrane due to the coffee stain effect, which was later reduced to AuNPs by thermal treatment of the films, as shown in Figure 10. The self-assembly of the metal precursor inside the pores followed by additional treatment allows the pore-selective functionalization of metal NPs which otherwise would have been difficult. Although this method successfully fabricates the NPs into the polymeric substrates, the current challenges include the incorporation of inorganic NPs (typically metals or metal oxides) embedded at the nanoscale in different organic structures such as polymers without any complications and multiple steps.

### 2.3. Pore-Selective Functionalization of HCP Films Accompanying Chemical Reaction

A new BF process accompanying in situ interfacial chemical reactions has been introduced as an emerging field for the pore-selective functionalization of HCP films. When we examine the BF mechanism in more detail, one can observe the formation of an aqueous droplet/polymer solution interface during the pore formation process. This interface was used for pore functionalization by the process of interfacial chemical reactions. The interfacial chemical reactions are performed by taking the reactants in two or more different immiscible phases and allowed to react at the interface, the reaction potential along with the concentration gradient continuously drives the reactants toward the interface, thereby carrying the reaction to completion or high product yield at the interface. The most common interfaces used in interfacial reactions are two immiscible liquids or solid and liquid [100]. In the BF process, an aqueous droplet/polymer solution interface can be found, as illustrated in Figure 11, which can be employed to perform interfacial chemical reactions for pore-selective functionalization of HCP films, i.e., functionalization could be obtained only in the pores of HCP film. The modified BF method is briefly described as follows: (a) polymer solution containing reactant A in a volatile solvent under humid conditions, (b) condensation of water droplets containing reactant B at the polymer surface; after the close packing of water droplets and self-assembly of reactant A around the water droplets, (c) formation of product C at the polymer solution/water droplet interface due to the interfacial reaction between reactant A and reactant B, and finally after the evaporation of water droplets and drying of the film (d) pore-functionalized HCP film was obtained, as shown in Figure 11.

For convenience, the functionalization methods of HCP films by interfacial chemical reactions were divided into two types based on the humid conditions, i.e., water as a reactant and water containing another material as a reactant.

#### 2.3.1. Pore-Functionalization by BF Process Using Water as a Reactant

For the method, a component reactive to water was added in the polymer solution such as titanium chloride [101,102], or tin chloride [103], and cast under humid conditions. This favored a chemical reaction to happen at the aqueous droplet/polymer solution interface and resulted in their respective products at the pores as shown in Figure 12. Li et al. fabricated composite film with hemispherical or mushroom-like titanium oxide (TiO_2_) microparticles lying in the pores of an HCP PS film which can be used with or without the polymer matrix, as shown in Figure 13. Similarly, Shin et al. reported the fabrication of PS HCP films with TiO_2_ NPs filled pores with slight modification in the BF process [104]. Here, the micro-water droplets were used as a reactive template, and titanium butoxide (Ti(C_4_H_9_O)_4_) was poured into the partially dried PS solution during the BF process. The (Ti(C_4_H_9_O)_4_) reacts with micro-water droplets and forms TiO_2_ NPs in the pores shown in Figure 14. In most of these reports, the aqueous droplets were used as micro-reactors for the preparation of metal hydroxides/oxides with specific shapes but not for the functionalization of pores in HCP film, given in Table 2. For example, using it as template for the fabrication of microdome selectively containing TiO_2_ NPs, which may have potential applications in different fields of electronic optoelectronic and anti-reflective surfaces [105]. In another example, water droplets were used as polymerization initiators for the polymerization of acrylate monomer to obtain polyacrylate HCP film [106]. The aggregation of polymer around the water droplets triggered the polymerization of alkylcyanoacrylate, and the –OH heads of the formed chains were anchored on the water droplets and the growing chains persisted at the water–solution interface forming a pore-selective HCP film, as shown in Figure 15. 

#### 2.3.2. BF Process Accompanying Chemical Reactions with a Non-Aqueous Reactant 

In the bottom-up process of the BF mechanism, the stage at which there is an interfacial interaction of water droplet and polymer solution can be applied for the site-selective functionalization. Applying this phenomenon, Huh et al. recently introduced a modified BF method, called the reactive BF (rBF) method, for the single-step fabrication of pore-selective functionalized films by accompanying an interfacial chemical reaction during the BF process [98,109,110,111,112,113]. For this method, a solvent-soluble reactant is added to the hydrophobic polymer solution, and a water-soluble counter reactant is added to the water used for humid conditions. When the polymer solution containing functionalizing reactant is cast under humid conditions containing counter reactant, a chemical reaction happens at the aqueous droplet/polymer solution interface and the product is formed in the pore of HCP film as a coating, summarized in Table 3 [109,110,111,112]. The benefit of this approach is that any material, either organic [109,110,111,112] or inorganic [113], can be functionalized by this method in one single step. Male et al. fabricated pore functionalized polyaniline (PANI) HCP porous films by in-situ polymerization via the BF method [110]. They added benzoyl peroxide to the PS solution and cast under reactive vapor of aniline hydrochloride, which resulted in PANI functionalized pores which otherwise might be impossible due to poor solubility of PANI in most of the solvents, as shown in Figure 16. Cao et al. fabricated a pore-selective carboxyl group functionalized PI HCP porous films using potassium hydroxide (KOH) as a counter reactant in humidity and decorated AgNPs at the pores, as shown in Figure 17 [111]. Similarly, tin sulfide (SnS) was functionalized in the pores of PS HCP porous films using H_2_S in the humidity as the reacting agent, shown in Figure 18 [109]. The pore-selective coating of SnS was used as a template for the fabrication of a moth-eye patterned film for antireflection due to its photo-responsive property under solar stimulated light illumination [114]. Modigunta et al. fabricated a temperature-sensitive polymer, poly (N-isopropyl acrylamide) (PNIPAAm), inside the pores of the HCP porous films using this same strategy. They oxidized the PS-CHO assembled inside the pores of HCP porous films using oxone to fabricate –COOH in the pores of the HCP porous films [112]. The oxidization of the aldehyde group occurred at the interface of the water droplet/polymer solution accompanying an interfacial reaction. This PS-pf-COOH film was functionalized with amine-terminated PNIPAAm (NH_2_-PNIPAAm) via EDC coupling as shown in Figure 19. Later, they showed capturing of Ag particles in the pores of the HCP film at the LCST of PNIPAAm. This opens a plethora of prospects to immobilize various types of functional or smart materials containing hydroxyl or amine groups. The fabrication of pore-selective –COOH functionalized HCP films by the rBF method may have various applications, such as in controlled drug release [115,116] or biosensors [117], because the –COOH group functionalized at the pores can be further immobilized with various types of functional or smart materials containing hydroxyl or amine groups [112,118]. Recently, Falak et al. fabricated HCP porous PS films with pore-selective Ag using the rBF method accompanied by an interfacial reaction, shown in Figure 20 [119]. In this study, an inorganic reactant AgNO_3_ was added in the aq. humidity as a functionalizing agent. On the other hand, ferrocene was included in the PS polymer solution as a reducing agent, which reduced AgNO_3_ to Ag inside the pores of the HCP films. This strategy might be useful for 3D micro-patterning of biological moieties of interest to the pores for applications in tissue engineering, antibacterial membranes, protein- and cell-based biosensors, microelectronic devices, or filtration membranes [119].

Since the discovery of the BF method in 1994, the past decade has seen rapid developments in the BF approach. HCP films fabricated via the BF method have shown great potential in chemistry and materials science. As the porous films are composed of the polymer framework and pores, their use can be applicable to frameworks and pores [121]. Compared with the vigorous progress of the framework, the applications of pores is less reported, acting as secondary templates [105,114,122], selective antibacterial surfaces [5,113,123], catalysis [70,71], selective cell culturing [5,123], drug capture and release [95,115], or in protein recognition [87,89], as shown in Figure 21. Thus, how to develop the new functions of the cavities becomes very meaningful, which will bring new vitality to the field of porous films.

## 3. Conclusions

This study has been one of the first attempts to thoroughly review the pore-selective functionalization of the HCP films via the modified BF methods. In this review, we have focused more on the use of rBF accompanying chemical reactions for the pore-selective functionalization of the HCP films. The functionalization of pore surfaces in one step is possible by using dynamic BF processes, in which there is an active flow of vapors over the surface of the polymeric solution facilitating the evaporation of water droplets containing the chemical moieties dissolved in it, thereby facilitating the interfacial reaction between the two reactants at the interface of the polymer/water droplet and functionalization of the pore surfaces. However, in the static BF process, the effects of vapor flow could be neglected due to the passive exposure of polymeric solutions without inducing extra vapor flow. Therefore, pore-selective functionalization using the self-assembly of amphiphilic polymers is feasible with the static BF method, while pore-selective functionalization using the self-assembly accompanying the interfacial chemical reaction between two reactants is not achievable. Earlier, optimizing versatility in functionalizing these pore surface structures required an ex situ process, and the in situ functionalization was limited to surfactants that are compatible with the BF formation. Therefore, the possibility of combining the BF method with interfacial chemical reactions might push forward the development of pore-selective functionalization in both fundamental and application aspects. Although the principle of the BF method looks very simple, optimizing the concentration of reactants for the formation of products at the pores without compromising the pattern remains a challenge. In the modified BF method, the interfacial interactions between water droplets containing a reactant and the polymer solution containing the counter reactant play a key role. Understanding the kinetics and conditions of the chemical reaction is one of the most important aspects of the process. Additionally, the selection of reactants concerning their nature, solubility, or other aspects will help fabricate HCP films with the desired functionalization in a more controllable way. The traces of reactant or the irrelevant byproducts left at the pores may also be unfavorable in this system. Moreover, the chemical reactions possible for the pore-selective functionalization are limited to the formation and application of the new products formed. Nevertheless, this review provides an insight for future research as the modified BF method is still in its preliminary stage. The information provided in this review can be used to develop targeted studies aimed at a particular application. For example, the fabrication of pore selective –COOH opens up a plethora of opportunities for further immobilization of smart or functional polymers, proteins, ligands, and biomolecules which can extend the applications of the HCP films fabricated via the modified BF method.

## Figures and Tables

**Figure 1 nanomaterials-12-01055-f001:**
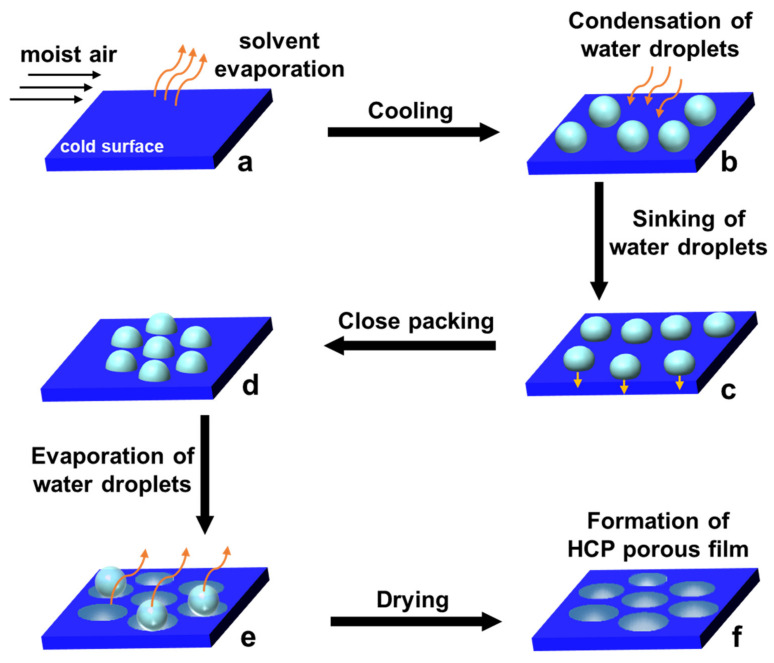
A sequence of stages for the preparation of HCP porous film by the breath-figure method. (**a**) Polymer solution in a volatile solvent under humid conditions; (**b**) condensation of water droplets due to cooling of surface temperature; (**c**) water droplets grow bigger and sink into the solution; (**d**) packing of water droplets; (**e**) arrangement of water droplets into ordered two- or three-dimensional (2D or 3D) arrays; and (**f**) formation of HCP film after drying.

**Figure 2 nanomaterials-12-01055-f002:**
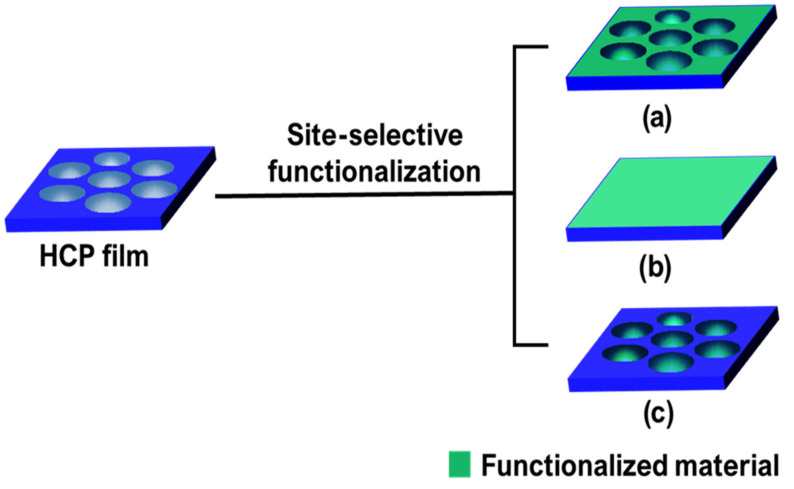
Possible site-selective functionalization of HCP films by modified BF method. (**a**) Top, (**b**) bottom, and (**c**) pore surface functionalization.

**Figure 3 nanomaterials-12-01055-f003:**
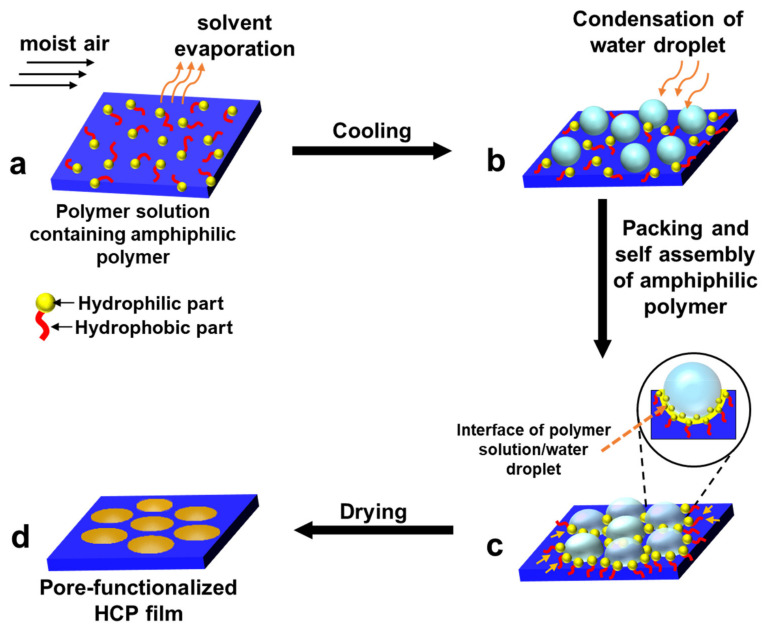
A sequence of stages of pore-selective functionalization by self-assembly of amphiphilic polymer. (**a**) Polymer solution containing amphiphilic polymer in a volatile solvent under humid conditions; (**b**) condensation of water droplets due to cooling of surface temperature; (**c**) close packing of water droplets and self-assembly of hydrophilic part of the amphiphilic polymer around the water droplets; (**d**) formation of pore-functionalized HCP film after drying.

**Figure 4 nanomaterials-12-01055-f004:**
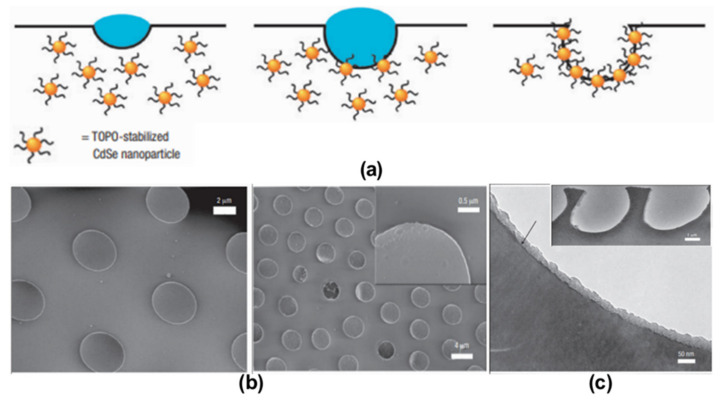
(**a**) Cross-sectional diagram of nanoparticle assembly at a water droplet–solution interface during the breath-figure formation. (**b**) SEM images of the surface of an NP-decorated BF film. (**c**) TEM images of cross-sections through porous PS film. Reprinted with permission from ref. [86]. Copyright 2004 Nature Materials.

**Figure 5 nanomaterials-12-01055-f005:**
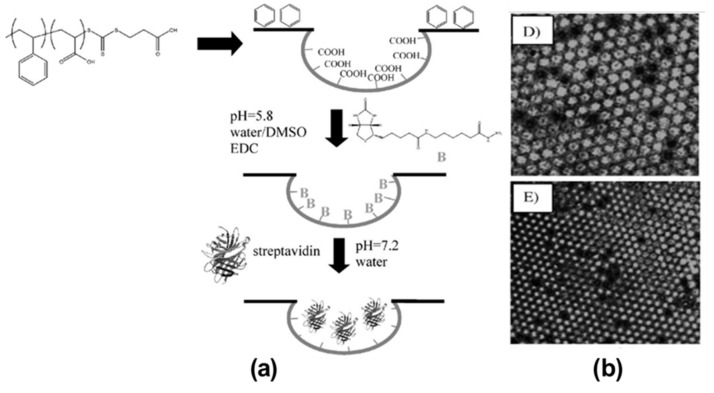
(**a**) Synthetic approach to a streptavidin microarray with immobilization of protein selectively in the pores. (**b**) Confocal microscopy images of PS-PAA honeycomb-structured porous films after modification with biotin and streptavidin. Reprinted with permission from ref. [87]. Copyright 2008 Advanced Materials.

**Figure 6 nanomaterials-12-01055-f006:**
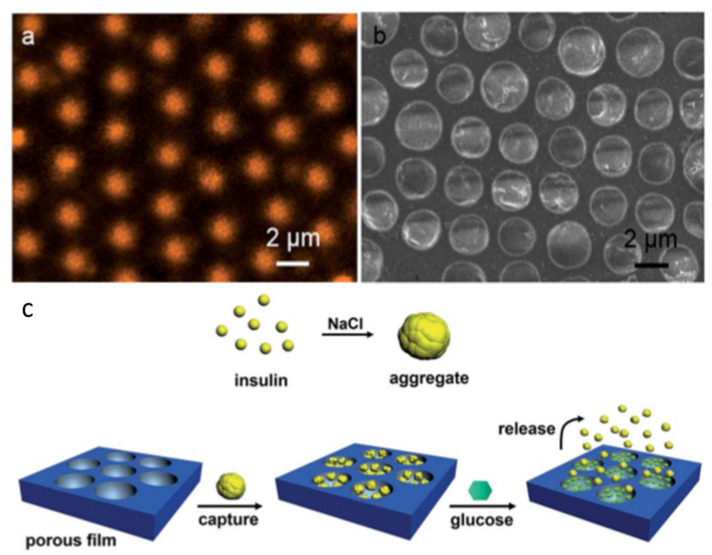
(**a**) CLSM images of PBA-PAA/PS porous film after their immersion in ARS dye solution. (**b**) SEM image of porous PS film loaded with insulin aggregates after immersion in 0.8 M NaCl aqueous solution at pH 5.3 at 25 °C. (**c**) Scheme of the capture of glucose and the glucose-responsive release of insulin aggregates. Reprinted with permission from ref. [95]. Copyright 2015 Journal of Materials Chemistry B.

**Figure 7 nanomaterials-12-01055-f007:**
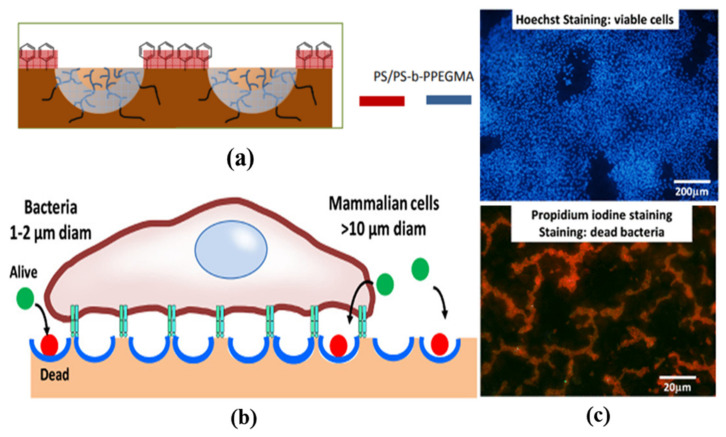
(**a**) Schematic representation of the chemical distribution of the blends of PS and PS-b-PPEGMA (75:25 wt% blend) at the surface. Reprinted with permission from ref. [80]. Copyright 2016 ACS Applied Materials & Interfaces. (**b**) Scheme of the strategy for selective cell culture. Mammalian cells interact only adhere to the top surface, while the bacteria may enter inside the pores due to size differences and get affected by antimicrobial functional groups. (**c**) Cell adhesion tests using (actin staining (red), and Hoechst (blue)) after 96 h of culture on the fabricated film. Reprinted with permission from ref. [5]. Copyright 2017 ACS Applied Materials & Interfaces.

**Figure 8 nanomaterials-12-01055-f008:**
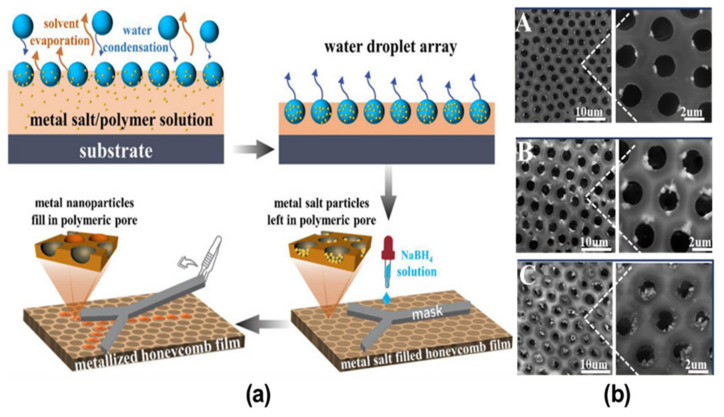
(**a**) The schematic representation showing the fabrication of the metal–NPs-functionalized honeycomb films. (**b**) SEM images of the HCP films filled with metal NPs: Au (A), Ag (B), and Cu (C). Reprinted with permission from ref. [97]. Copyright 2021 Polymers.

**Figure 9 nanomaterials-12-01055-f009:**
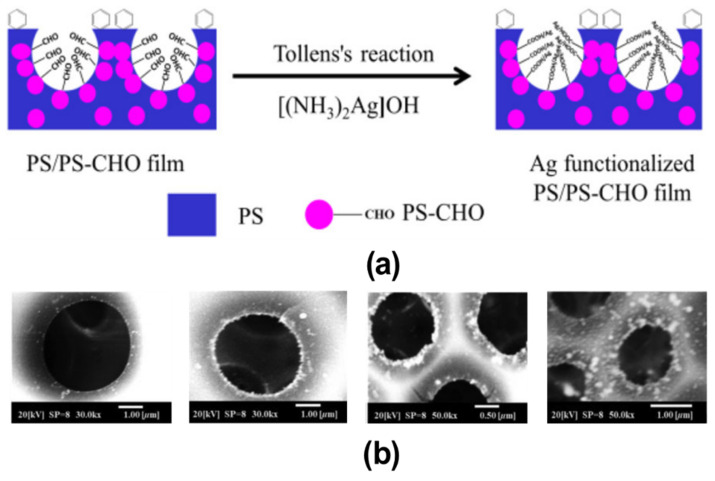
(**a**) Schematic representation of reaction mechanism of Tollens’s reagent with aldehyde group in HCP porous films fabricated using PS and PS-CHO blended solutions. (**b**) The typical SEM images of PS/PS-CHO HCP porous films with Ag functionalization after treating with Tollens’s reagent. Reprinted with permission from ref. [98]. Copyright 2018 Polymer Physics.

**Figure 10 nanomaterials-12-01055-f010:**
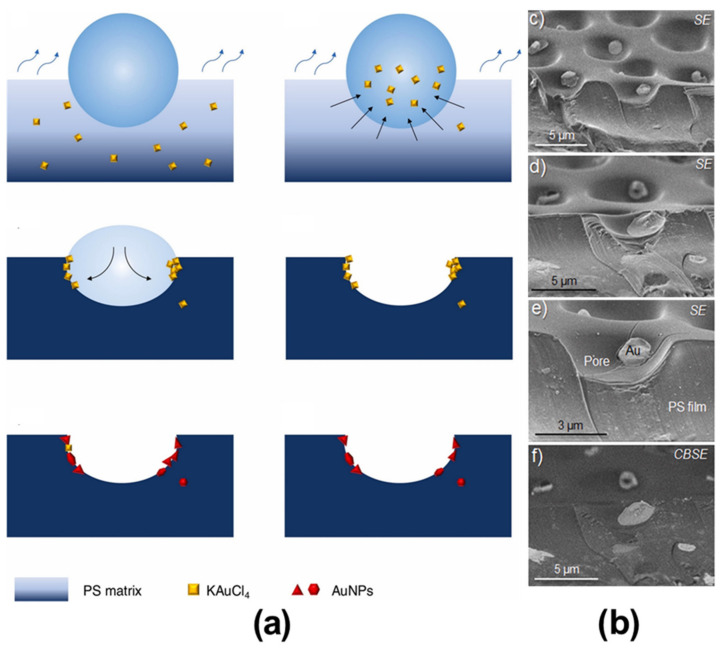
(**a**) Proposed formation mechanism of hybrid membranes with AuNPs inside the pores of the HCP film. (**b**) SEM analysis of sample Au5 after heating at 170 °C for 5 min, showing AuNPs present at the pores. Reprinted with permission from ref. [70]. Copyright 2021 Colloids and Surfaces A: Physicochemical and Engineering Aspects.

**Figure 11 nanomaterials-12-01055-f011:**
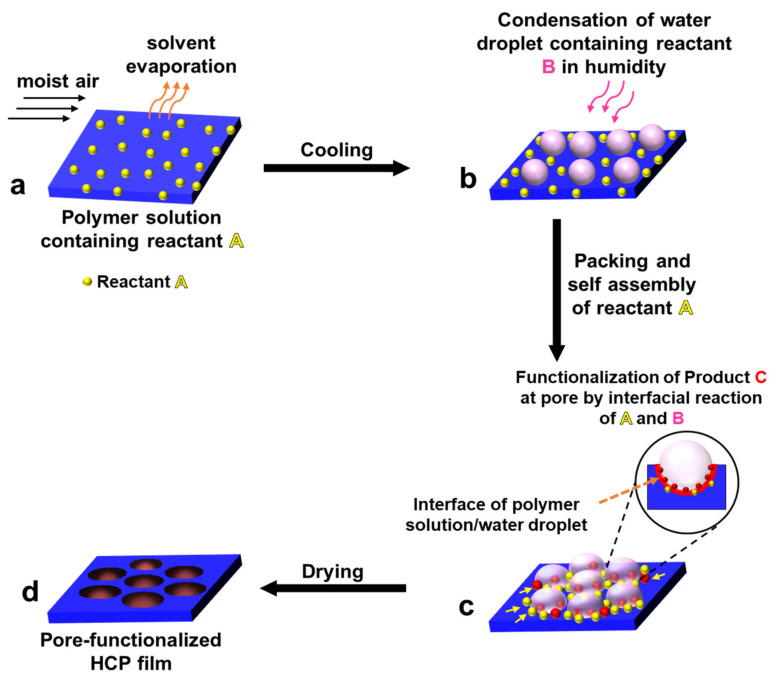
A sequence of stages of pore-selective functionalization by self-assembly accompanied by chemical reaction. (**a**) polymer solution containing reactant A, (**b**) condensation of water droplets containing counter reactant B, (**c**) functionalizing of product C at the pore by the interfacial reaction of reactant A and reactant B, (**d**) formation of pore-functionalized HCP film after drying.

**Figure 12 nanomaterials-12-01055-f012:**
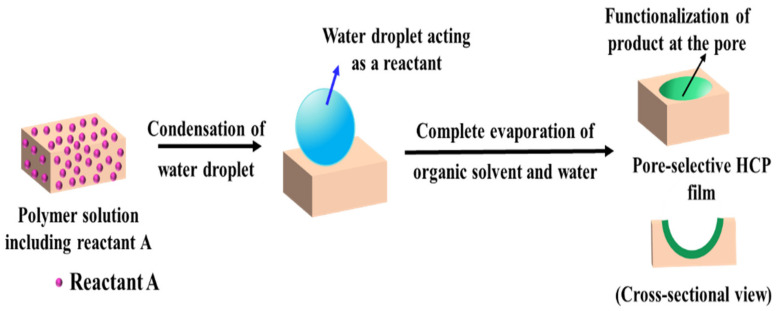
Pore-selective functionalization of HCP films by using water as a reactant. The polymer solution containing a reactant reactive to water is cast under humid conditions for the reaction to occur at the interface of polymer solution/water droplet to achieve pore-selective functionalization.

**Figure 13 nanomaterials-12-01055-f013:**
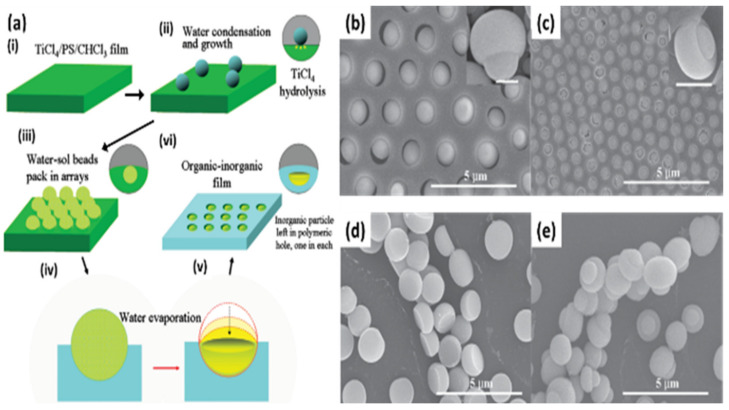
(**a**) Scheme for the formation of highly ordered asymmetrical inorganic particle/polymer composite films by BFs method. (**b**,**c**) SEM images of the obtained composite film from the TiCl_4_/PS/CHCl_3_ solution with different concentrations of TiCl_4_, SEM images of (**c**) hemispherical and (**d**,**e**) mushroom-like TiO_2_ microparticles after calcination of the composite film at 450 °C for 3 h. Reprinted with permission from ref. [102]. Copyright 2011 Journal of the American Chemical Society.

**Figure 14 nanomaterials-12-01055-f014:**
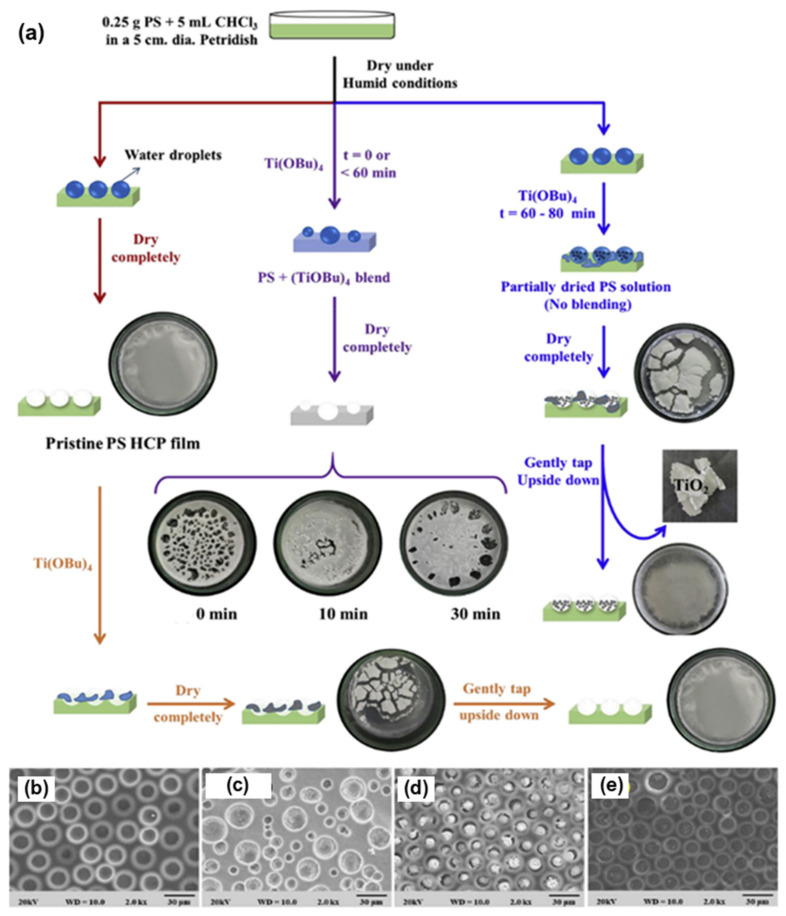
(**a**) Schematic representation showing different reaction conditions with the variance in time intervals of pouring (Ti(C_4_H_9_O)_4_) solution to the PS solution. (**b**–**e**) shows the typical SEM images of PS film obtained at a different time interval. Reprinted with permission from ref. [104]. Copyright 2018 Polymer.

**Figure 15 nanomaterials-12-01055-f015:**
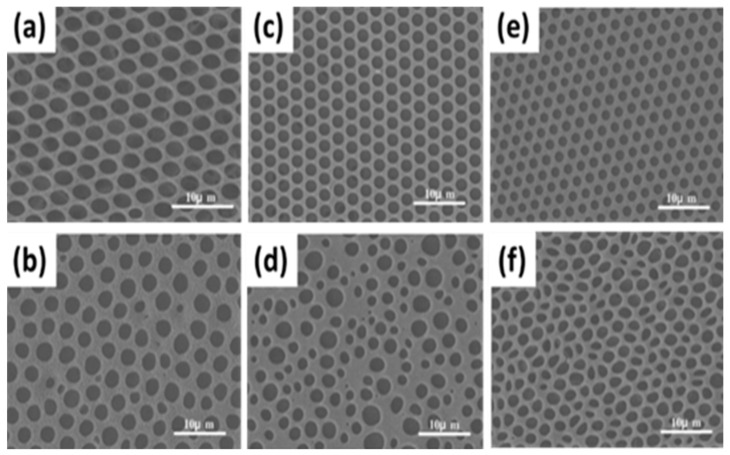
SEM images of porous PACA films fabricated from monomer solutions of (**a**) ECA, (**c**) BCA, (**e**) OCA, and polymer solutions of (**b**) PECA, (**d**) PBCA, (**f**) POCA. Reprinted with permission from ref. [106]. Copyright 2010 Journal of Colloid and Interface Science.

**Figure 16 nanomaterials-12-01055-f016:**
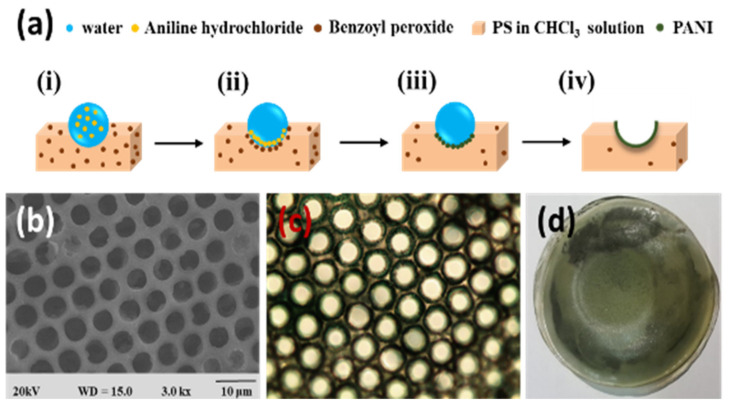
(**a**) Mechanism showing the formation of PANI-f-PS film with PANI functionalized pores: (i) Condensed aq. aniline hydrochloride droplets on PS-BPO solution during BF process, (ii) assembly of aniline hydrochloride and BPO at aqueous droplet/polymer solution interface, (iii) interfacial polymerization, (iv) and formation of PANI functionalized pores after completion of reactive BF process. (**b**) SEM, (**c**) optical, and (**d**) digital photograph of PANI-f-PS film. Reprinted with permission from ref. [110]. Copyright 2017 Polymer.

**Figure 17 nanomaterials-12-01055-f017:**
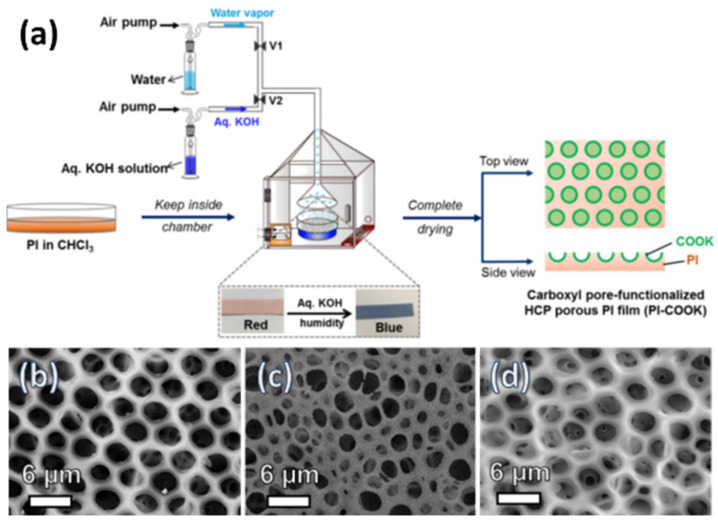
(**a**) Schematic representation showing the fabrication of the pore-selective carboxyl functionalized PI HCP films (PI-COOK) via BF method under aq. KOH humidity with a time interval after using initial water humidity. SEM images of PI films fabricated under (**b**) water humidity, (**c**) aq. KOH humidity, and (**d**) under aq. KOH humidity for 20 min after initial water humidity. Reprinted with permission from ref. [111]. Copyright 2018 Polymer.

**Figure 18 nanomaterials-12-01055-f018:**
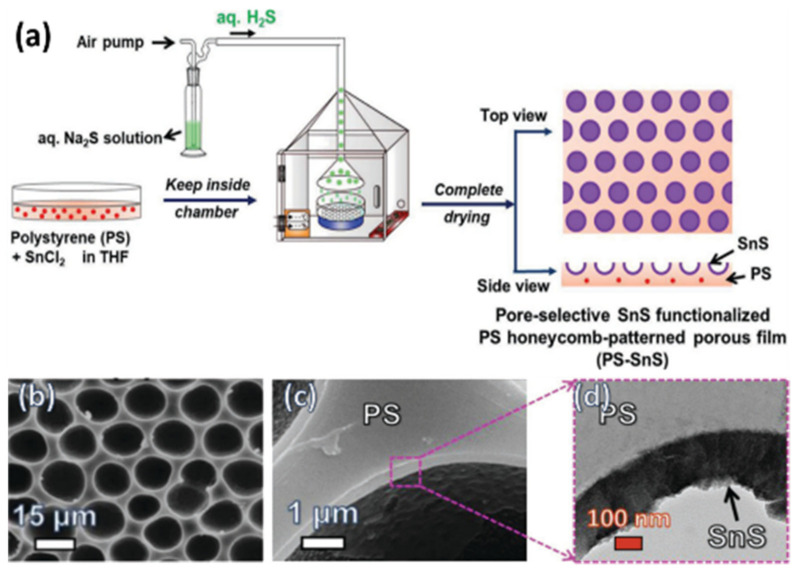
(**a**) Schematic representation showing the experimental process for the pore-selective SnS-functionalized PS HCP film using a modified BF method. SEM images of PS HCP films fabricated by casting PS solution containing SnCl_2_ under (**b**,**c**) aq. H_2_S humidity, and (**d**) magnified TEM image showing a thin layer of SnS. Reprinted with permission from ref. [109]. Copyright 2018 Advanced Materials Interfaces.

**Figure 19 nanomaterials-12-01055-f019:**
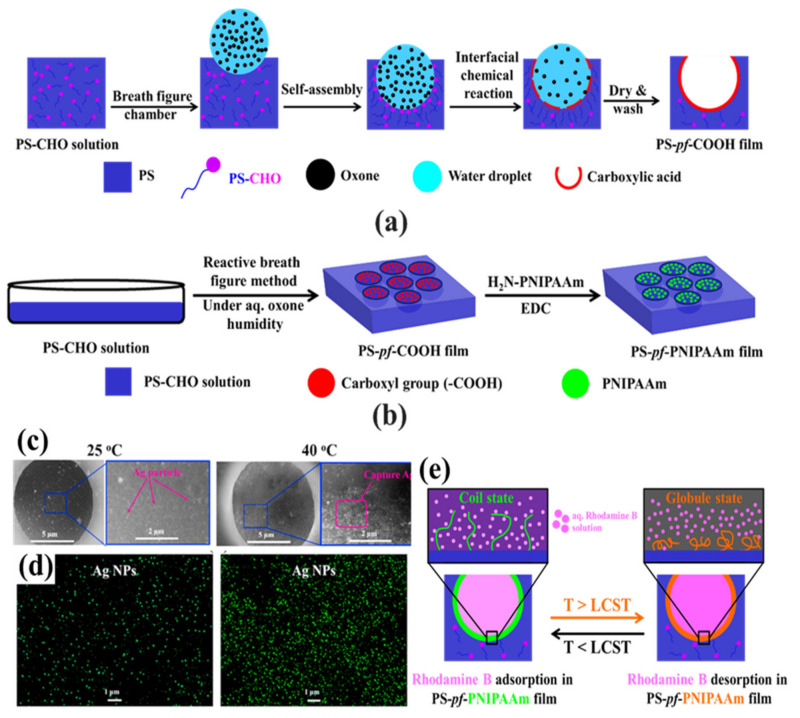
(**a**) The possible mechanism involved in the rBF process via self-assembly process accompanying an interfacial chemical reaction between formylated PS and oxone at the interface of the water droplet/polymer solution. (**b**) Schematic process for the preparation of pore-selective PNIPAAm modified (PS-pf-PNIPAAm) film by the reaction between PS-pf-COOH film and an aq. H_2_N-PNIPAAm solution in the presence of EDC. (**c**,**d**) SEM images and elemental mapping show the capture of Ag particles at 25 °C and 40 °C. Reprinted with permission from ref. [112] Copyright 2020 Polymer. (**e**) Scheme showing the release of rhodamine B at the LCST of PNIPAAm. Reprinted with permission from ref. [115] Copyright 2021 Polymer Bulletin.

**Figure 20 nanomaterials-12-01055-f020:**
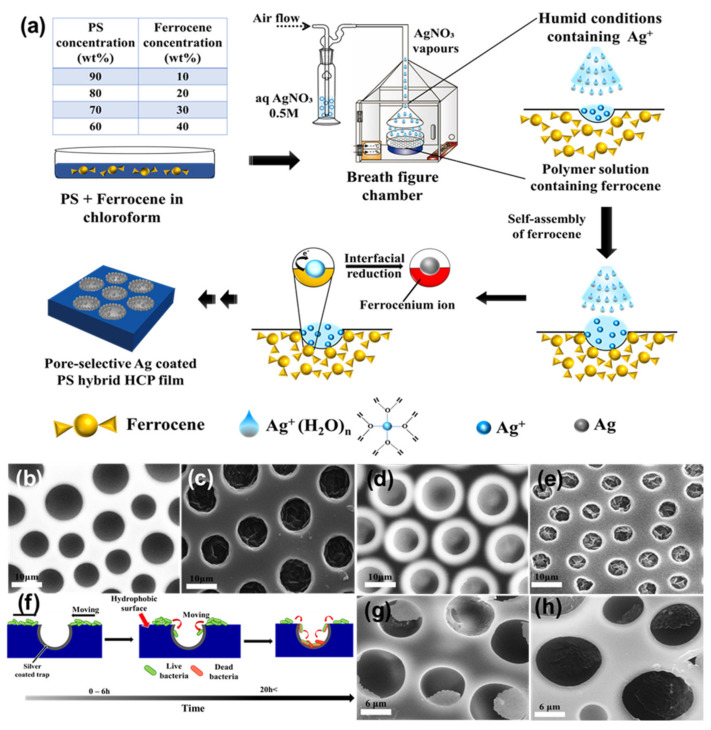
(**a**) Schematic representation showing the fabrication of the pore-selective Ag functionalized PS HCP films via rBF method under aq. AgNO_3_ humidity. Typical SEM images of the 10 and 20 wt% PS HCP films fabricated under (**b**,**d**) aq. humidity, and (**c**,**e**) aq. AgNO_3_ humidity, respectively. (**f**) The mechanism for trapping of bacteria inside the pits of the Ag-functionalized porous HCP film. SEM images of (**g**) *E. coli* and (**h**) *S. aureus* showing the trapped bacteria at the pores of the HCP film. Reprinted with permission from ref. [119] Copyright 2022 Polymer.

**Figure 21 nanomaterials-12-01055-f021:**
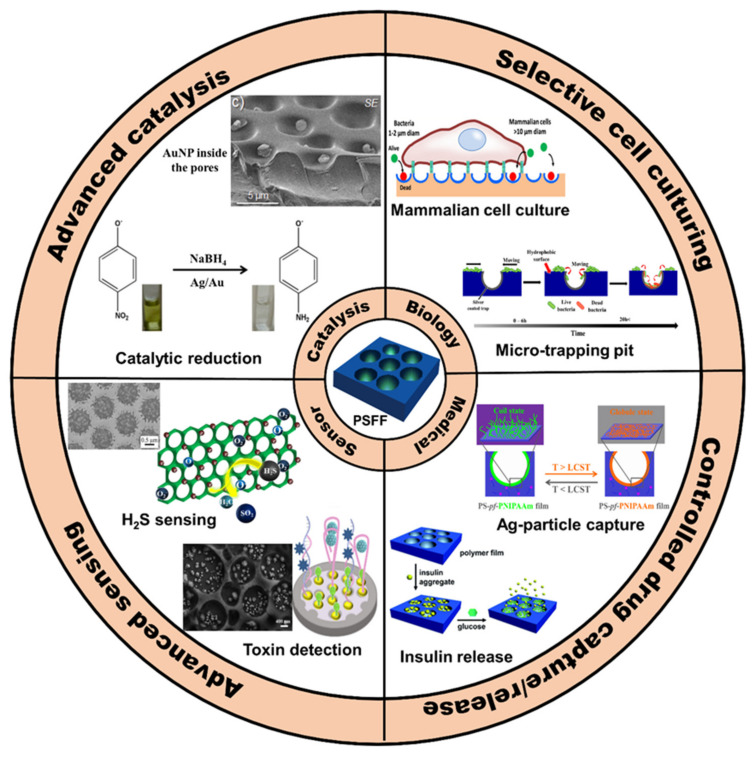
Pore-selectively functionalized HCP films via the modified BF method and their applications in biology, medical, sensor application, and catalysis. PSFF—Pore-Selectively Functionalized Film.

**Table 1 nanomaterials-12-01055-t001:** Summary of reports for the pore-selective functionalization of HCP films with a reactant in a polymer solution and using an external agent for completion of the reaction.

Reactant A	Treatment	Product at Pore	Ref.
HAuCl_4_, AgNO_3_, and CuCl_2_	NaBH_4_	Au, Ag, and Cu NPs	[97]
PS-CHO	Tollen’s Reagent	Ag	[98]
KAuCl_4_	Heat	Au NPs	[70]

**Table 2 nanomaterials-12-01055-t002:** Summary of reports using one reactant in polymer solution and water from the humid atmosphere as another reactant.

Reactant A in the Polymer Solution	Humid Conditions	Product/Functionalization in Pore	Ref.
Titanium tetrachloride	Water	Titanium dioxide	[101]
Titanium tetrachloride	Water	Titanium dioxide	[102]
Tin tetrachloride	Water	Tin dioxide	[103]
Titanium butoxide	Water	Titanium dioxide NPs	[104]
Titanium n-butoxide	Water	Titanium dioxide	[107]
Titanium tetraisopropoxide	Water	Titanium dioxide	[108]
Alkylcyanoacrylate	Water	PACA	[106]

**Table 3 nanomaterials-12-01055-t003:** Summary of reports using two reactants, one in the polymer solution and the other in a humid atmosphere for the functionalization of pores in HCP films.

Reactant A in the Polymer Solution	Reactant B in Humid Conditions	Functionalization in Pore (Product C)	Ref.
Poly (4-vinylpyridine)	Formic acid	PVP-FA precipitate	[120]
Benzoyl peroxide	Aniline	Polyaniline	[110]
Tin dichloride	Hydrogen sulfide	Tin sulfide	[109]
Polyimide	Potassium hydroxide	Carboxylic group	[111]
PS aldehyde (PS-CHO)	Oxone	Carboxylic acid (–COOH)	[112]
Ferrocene	Silver Nitrate	Silver	[119]

## Data Availability

Not applicable.

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
