# Peer review of "Modified Breath Figure Methods for the Pore-Selective Functionalization of Honeycomb-Patterned Porous Polymer Films"

_nanomaterials, 2022, doi:10.3390/nano12071055_

Round 1

Reviewer 1 Report

The manuscript of Falak and Huh reviews the recent development carried out in the field of Breath Figures method to produce honeycomb structured films. This review is mainly focused on strategies to selective functionalize the pores at the surface, and more specifically they review the approaches based on reactive breath figure process in which a chemical reaction takes place at the interface of the pore during the breath figures. These strategies are new, and have not been reviewed before. The review is well written and structured. I recommend its publication after minor revision.

Line 234. De León et al. instead of Alberto et al.

I suggest to move the paragraph between line 302-307 related the polymerization of alkylcyanoacrylate at the end of the section 2.3.1. This work is in the middle to discussion related to the formation of Titanium based materials inside the pores and I recommend to move after the description of all the materials related to titanium and tin inorganic materials. Likewise, in Table 2, it would be better to put reference 102 in the last line of the table.

I also suggest to delete Figure e) from Figure 18 as there is no explanation concerning the preparation of such structures

Author Response

The manuscript of Falak and Huh reviews the recent development carried out in the field of Breath Figures method to produce honeycomb structured films. This review is mainly focused on strategies to selective functionalize the pores at the surface, and more specifically they review the approaches based on reactive breath figure process in which a chemical reaction takes place at the interface of the pore during the breath figures. These strategies are new, and have not been reviewed before. The review is well written and structured. I recommend its publication after minor revision.

Thank you for your encouraging comments.

1. Line 234. De León et al. instead of Alberto et al.

R1. Thank you for your comment. We have changed Alberto et al. to De León et al.

2. I suggest to move the paragraph between line 302-307 related the polymerization of alkylcyanoacrylate at the end of the section 2.3.1. This work is in the middle to discussion related to the formation of Titanium based materials inside the pores and I recommend to move after the description of all the materials related to titanium and tin inorganic materials. Likewise, in Table 2, it would be better to put reference 102 in the last line of the table.

R2. Thank you for your valuable comment. The paragraph between line 302-307 has been moved to the end of the section 2.3.1. as follows:

 “Similarly, Shin et al. reported the fabrication of PS HCP films with TiO2 NPs filled pores with slight modification in the BF process. [102] Here, the micro-water droplets were used as a reactive template, and titanium butoxide (Ti(C4H9O)4) was poured into the partially dried PS solution during the BF process. The (Ti(C4H9O)4) reacts with micro-water droplets and forms TiO2 NPs in the pores shown in Figure 14. In most of these reports, the aqueous droplets were used as micro-reactors for the preparation of metal hydroxides/oxides with specific shapes but not for the functionalization of pores in HCP film, given in Table 2. Like, using it as a template for the fabrication of microdome selectively containing TiO2 NPs which may have potential applications in different fields of electronic optoelectronic and anti-reflective surfaces. [103] In another example, water droplets were used as polymerization initiators for the polymerization of acrylate monomer to obtain polyacrylate HCP film. [104] The aggregation of polymer around the water droplets triggered the polymerization of alkylcyanoacrylate, and the –OH heads of the formed chains were anchored on the water droplets and the growing chains persisted at the water–solution interface forming a pore-selective HCP film as shown in Figure 15.

Likewise, the figure number and references have been adjusted. The polymerization of alkylcyanoacrylate has also been moved to the bottom of Table 2 as follows:

Reactant A in the Polymer solution

Humid conditions

Product/Functionalization in pore

Ref.

Titanium tetrachloride

Water

Titanium dioxide

[99]

Titanium tetrachloride

Water

Titanium dioxide

[100]

Tin tetrachloride

Water

Tin dioxide

[101]

Titanium butoxide

Water

Titanium dioxide NPs

[102]

Titanium n-butoxide

Water

Titanium dioxide

[105]

Titanium tetraisopropoxide

Water

Titanium dioxide

[106]

Alkylcyanoacrylate

Water

PACA

[104]

3. I also suggest to delete Figure e) from Figure 18 as there is no explanation concerning the preparation of such structures.

R3. Thank you for your valuable comment. Figure (e) from Figure 18 has been removed as advised. 

Reviewer 2 Report

Review attached

Author Response

Modified breath figure method for the pore-selective functionalization of honeycomb-patterned porous polymer films

Shahkar Falak 1, and Do Sung Huh.

This review is discussing different breath-figure strategies employed in the creation of polymer

films with honeycomb patterns. Materials of that type are attracting interest due to their broad

application potential and therefore this review might provide useful additional information for

the peculiarities of the process and emerging usage.

Manuscripts of that type should be evaluated towards three main criteria:

1) Comprehensive coverage of the existing literature including the newest developments.

The authors are very close in fulfilling this requirement, and are advised to change/replace some non-relevant articles and update with some recent publications:

(a) reference 31 does not report dendrimers for BF arrays, please replace with - Luke A. Connal, L. A. et al. Dramatic Morphology Control in the Fabrication of Porous Polymer Films. Adv. Funct. Mater. 2008, 18, 3706-3714;

R(a). Thank you for your comment. The previous reference 31 has been replaced as advised.

  1. Connal, L.A.; Vestberg, R.; Hawker, C.J.; Qiao, G.G. Dramatic morphology control in the fabrication of porous polymer films. Adv. Funct. Mater. 2008, 18, 3706-3714.

(b) please add the following paper to references 37–41, it represents a different class of copolymer – Liu, X. et al. Controlled ATRP Synthesis of Novel Linear-Dendritic Block Copolymers and Their Directed Self Assembly in Breath Figure Arrays. Polymers 2019, 11(3), 539;

R(b). Thank you for your comment. The above reference has been added to references 27-41 as advised.

  1. Liu, X.; Monzavi, T.; Gitsov, I. Controlled ATRP synthesis of novel linear-dendritic block copolymers and their directed self-assembly in breath figure arrays. Polymers 2019, 11, 539.

(c) please, add the following review to the section references at the bottom of page 19, lines 431-440 –Chen, S.; Gao, S.; Jing, J.; Lu, Q. Designing 3D Biological Surfaces via the Breath Figure Method. Adv. Health Mater. 2018, 7(6), 1701043.

R(c). The above reference has been added as follows:

“As the porous films are composed of the polymer framework and pores, their use can be applicable to frameworks and pores. [121]”

  1. Chen, S.; Gao, S.; Jing, J.; Lu, Q. Designing 3D Biological Surfaces via the Breath‐Figure Method. Adv. Healthc. Mater. 2018, 7, 1701043.

2) Adequate discussion of the main advantages and disadvantages of the existing methods.

While the length of the text is acceptable for the scope of the review, these elements are generally missing throughout the body of the manuscript. It would be beneficial to see a critical comparison between the two major mechanisms of the breath figure formation: active flow of the aqueous vapors over the surface of the polymeric solution vs passive exposure.

R2. Thank you for your valuable comment. The advantages and disadvantages of the two methods with respect to the dynamic and static BF processes have been included in the conclusion as follows:

“The functionalization of pore surfaces in one-step is possible by using the dynamic BF processes, in which there is an active flow of vapors over the surface of the polymeric solution facilitating the evaporation of water droplets containing the chemical moieties dissolved in it. However, in the static BF process, the effects of vapor flow could be neglected due to the passive exposure of polymeric solutions without inducing extra vapor flow. Therefore, pore-selective functionalization using the self-assembly of amphiphilic polymers is feasible with the static BF method while with the rBF method is not achievable.”

3) Clarity of presentation.

The language is mostly clear with few typographical errors. Some required editing is mentioned in the technical suggestion below.

Technical

  1. The title of the paper needs small editing: “method” (singular) should be written “methods” (plural) since more than one method is reviewed.

R1. Thank you so much for your valuable comments. We have edited the title and it is as follows:

“Modified breath figure methods for the pore-selective functionalization of honeycomb-patterned porous polymer films”

  1. Include the legends and titles of Figure S1, Table S1 and Video S1.

R2. We do not have any Supplementary Materials. The statement “Supplementary Materials: The following supporting information can be downloaded at: www.mdpi.com/xxx/s1, Figure S1: title; Table S1: title; Video S1: title” is from the Journal’s template file. We have edited the statement as follows to avoid any confusion:

Supplementary Materials: No supplementary materials.”